# Dimension-Free Empirical Entropy Estimation

**Doron Cohen**
Department of Computer Science
Ben-Gurion University of the Negev
Beer-Sheva, Israel
doronv@post.bgu.ac.il

**Aryeh Kontorovich**
Department of Computer Science
Ben-Gurion University of the Negev
Beer-Sheva, Israel
karyeh@cs.bgu.ac.il

**Aaron Koolyk**
Department of Computer Science
Hebrew University
Jerusalem, Israel
aaron.koolyk@mail.huji.ac.il

**Geoffrey Wolfer**
JSPS International Research Fellow
Department of Computer
and Information Sciences
Tokyo University of Agriculture
and Technology
Tokyo, Japan [*]
geo-wolfer@m2.tuat.ac.jp

## Abstract

We seek an entropy estimator for discrete distributions with fully empirical accuracy bounds. As stated, this goal is infeasible without some prior assumptions on the distribution. We discover that a certain information moment assumption renders the problem feasible. We argue that the moment assumption is natural and, in some sense, *minimalistic* — weaker than finite support or tail decay conditions. Under the moment assumption, we provide the first finite-sample entropy estimates for infinite alphabets, nearly recovering the known minimax rates. Moreover, we demonstrate that our empirical bounds are significantly sharper than the state-of-the-art bounds, for various natural distributions and non-trivial sample regimes. Along the way, we give a dimension-free analogue of the Cover-Thomas result on entropy continuity (with respect to total variation distance) for finite alphabets, which may be of independent interest.

## 1 Introduction

Estimating the entropy of a discrete distribution based on a finite iid sample is a classic problem with theoretical and practical ramifications. Considerable progress has been made in the case of a finite alphabet, and the countably infinite case has also attracted a fair amount of attention in recent years. A less-addressed issue is one of *empirical* accuracy estimates: data-dependent bounds adaptive to the particular distribution being sampled.

Our point of departure is the simpler (to analyze) problem of estimating a discrete distribution $\boldsymbol{\mu}$ in total variation norm $\|\cdot\|_{\mathsf{TV}} = \frac{1}{2}\|\cdot\|_1$, where the most recent advance was made by Cohen et al. [2020]; see therein for a literature review. If $\boldsymbol{\mu}$ is a distribution on $\mathbb{N}$ and $\hat{\boldsymbol{\mu}}_n$ is its empirical realization based on a sample of size $n$, then Theorem 2.1 of Cohen et al. states that with probability at least $1 - \delta$,

$$\|\boldsymbol{\mu} - \hat{\boldsymbol{\mu}}_n\|_1 \quad \leq \quad \frac{2}{\sqrt{n}}\sum_{j\in\mathbb{N}}\sqrt{\hat{\boldsymbol{\mu}}_n(j)} + 6\sqrt{\frac{\log(2/\delta)}{2n}}. \tag{1}$$

---

[*]A major part of this research was conducted when the author was graduate student at Ben-Gurion University of The Negev, Israel.

35th Conference on Neural Information Processing Systems (NeurIPS 2021).

This bound has the advantage of being valid for all distributions on $\mathbb{N}$, without any prior assumptions, and being fully empirical: it yields a risk estimate that is computable based on the observed sample, not depending on any unknown quantities. (Additionally, Cohen et al. argue that (1) is near-optimal in a well-defined sense.) The question we set out to explore in this paper is: What analogues of (1) are possible for discrete entropy estimation?

When $\boldsymbol{\mu}$ has support size $d < \infty$, an answer to our question is readily provided by combining (1) with Cover and Thomas [2006, Theorem 17.3.3], which asserts that, for $\|\boldsymbol{\mu} - \boldsymbol{\nu}\|_1 \leq 1/2$, we have

$$|\mathrm{H}(\boldsymbol{\mu}) - \mathrm{H}(\boldsymbol{\nu})| \quad \leq \quad \|\boldsymbol{\mu} - \boldsymbol{\nu}\|_1 \log \frac{d}{\|\boldsymbol{\mu} - \boldsymbol{\nu}\|_1}, \tag{2}$$

where $\mathrm{H}(\cdot)$ is the entropy functional defined in (3). Indeed, taking $\boldsymbol{\mu}$ as in (1) and $\boldsymbol{\nu}$ to be $\hat{\boldsymbol{\mu}}_n$ yields a fully empirical estimate on $|\mathrm{H}(\boldsymbol{\mu}) - \mathrm{H}(\hat{\boldsymbol{\mu}}_n)|$. For fixed $d < \infty$, no technique relying on the plug-in estimator can yield minimax rates [Wu and Yang, 2016]. The plug-in is, however, minimax optimal for fixed $d < \infty$ [Paninski, 2003] as well as strongly universally consistent even for $d = \infty$ [Antos and Kontoyiannis, 2001a], and is among the few methods for which explicitly computable finite-sample risk bounds are known.

The thrust of this paper is to replace the restrictive finite-support assumption with considerably more general moment conditions. It is well-known that when estimating the mean of some random variable $X$, the first-moment assumption $\mathbb{E}|X| \leq M$ is not sufficient to yield any finite-sample information.[2] Strengthening the assumption to $\mathbb{E}|X|^\alpha \leq M$, for any $\alpha > 1$, immediately yields finite-sample empirical estimates on $\left|\mathbb{E}X - \frac{1}{n}\sum_{i=1}^n X_i\right|$ via the von Bahr and Esseen [1965] inequality.[3] In this sense, a bound on the $(1 + \varepsilon)$th moment is a *minimal* requirement for empirical mean estimation. However, it is not immediately obvious how to apply this insight to the entropy estimation problem: the corresponding random variable is $X = -\log \boldsymbol{\mu}(I)$, where $I \sim \boldsymbol{\mu}$, but rather than being given iid samples of $X$, we are only given draws of $I$.

**Our contribution.** In Theorem 1, we provide a dimension-free analogue of (2), which, combined with (1), allows for empirical accuracy bounds on the plug-in entropy estimator under a minimalistic moment assumption. Moreover, for this rich class of distributions, the plug-in estimator turns out to be asymptotically optimal, as we show in Theorem 4. Our moment assumption is natural and essentially the weakest one that makes *any* empirical bounds feasible, as we argue in Theorem 3. As we demonstrate in Section 6, the rates provided by our empirical bound compare favorably against the state of the art.

## 2 Definitions and notation

Our logarithms will always be base $e$ by default. For discrete distributions, there is no loss of generality in taking the domain to be the natural numbers $\mathbb{N} = \{1, 2, 3, \ldots\}$. For $k \in \mathbb{N}$, we write $[k] := \{i \in \mathbb{N} : i \leq k\}$. The set of all probability distributions on $\mathbb{N}$ will be denoted by $\Delta_{\mathbb{N}}$. For $d \in \mathbb{N}$, we write $\Delta_d \subset \Delta_{\mathbb{N}}$ to denote those $\boldsymbol{\mu}$ whose support is contained in $[d]$.

We define the operator $(\cdot)^{\downarrow}$, which maps any $\boldsymbol{\mu} \in \Delta_{\mathbb{N}}$ to its non-increasing rearrangement $\boldsymbol{\mu}^{\downarrow}$. The set of all non-increasing distributions will be denoted by $\Delta_{\mathbb{N}}^{\downarrow} := \{\boldsymbol{\mu}^{\downarrow} : \boldsymbol{\mu} \in \Delta_{\mathbb{N}}\}$.

We write $\mathbb{R}_+ := [0, \infty)$. For any $\xi : \mathbb{N} \to \mathbb{R}_+$ and $\alpha \geq 0$, define

$$\mathrm{H}^{(\alpha)}(\xi) := \sum_{j \in \mathbb{N} : \xi(j) > 0} \xi(j) |\log \xi(j)|^\alpha. \tag{3}$$

For $\xi \in \mathbb{R}^{\mathbb{N}}$, denote by $|\xi| \in \mathbb{R}_+^{\mathbb{N}}$ the elementwise application of $|\cdot|$ to $\xi$. When $\xi \in \Delta_{\mathbb{N}}$ and $\alpha = 1$, (3) recovers the standard definition of entropy, which we denote by $\mathrm{H}(\xi) := \mathrm{H}^{(1)}(\xi)$. For general

---

[2]Even distinguishing, for $X \geq 0$, between $\mathbb{E}X = 0$ and $\mathbb{E}X = M$ based on a finite sample is impossible with any degree of confidence. Of course, $\frac{1}{n}\sum_{i=1}^n X_i \to \mathbb{E}X$ almost surely, by the strong law of large numbers.

[3]Put $Y = X - \mathbb{E}X$; then $\mathbb{E}|Y| \leq 2M$. For $1 < \alpha < 2$, a sharper version of the Bahr-Esseen inequality [Pinelis, 2015] states that $\mathbb{E}\left[\left|\sum_{i=1}^n Y_i\right|^\alpha\right] \leq 2n(2M)^\alpha$, which implies tail bounds via Markov's inequality. Better rates are available via the median-of-means estimator, see Lugosi and Mendelson [2019].

$\alpha > 0$, this quantity may be referred to as the $\alpha$th *moment of information*. For $h \geq 0$, define

$$\Delta_{\mathbb{N}}^{(\alpha)}[h] = \left\{ \boldsymbol{\mu} \in \Delta_{\mathbb{N}} : \mathrm{H}^{(\alpha)}(\boldsymbol{\mu}) \leq h \right\}$$

and also $\Delta_{\mathbb{N}}^{(\alpha)} := \bigcup_{h \geq 0} \Delta_{\mathbb{N}}^{(\alpha)}[h]$ and $\Delta_{\mathbb{N}}^{\downarrow(\alpha)}[h] := \Delta_{\mathbb{N}}^{\downarrow} \cap \Delta_{\mathbb{N}}^{(\alpha)}[h]$.

For $n \in \mathbb{N}$ and $\boldsymbol{\mu} \in \Delta_{\mathbb{N}}$, we write $\boldsymbol{X} = (X_1, \ldots, X_n) \sim \boldsymbol{\mu}^n$ to mean that the components of the vector $\boldsymbol{X}$ are drawn iid from $\boldsymbol{\mu}$. The empirical measure $\hat{\boldsymbol{\mu}}_n \in \Delta_{\mathbb{N}}$ induced by the sample $\boldsymbol{X}$ is defined by $\hat{\boldsymbol{\mu}}_n(j) = \frac{1}{n} \sum_{i \in [n]} \mathbf{1}[X_i = j]$. For any $\xi \in \mathbb{R}^{\mathbb{N}}$ and $0 < p < \infty$, the $\ell_p$ (pseudo)norm is defined by $\|\xi\|_p^p = \sum_{j \in \mathbb{N}} |\xi(j)|^p$ and $\|\xi\|_\infty = \sup_{j \in \mathbb{N}} |\xi(j)|$.

For $\alpha, h > 0$, and $n \in \mathbb{N}$, define the $L_1$ *minimax risk* for the $\alpha$th moment by

$$\mathcal{R}_n^{(\alpha)}(h) \quad := \quad \inf_{\hat{H}} \sup_{\boldsymbol{\mu} \in \Delta_{\mathbb{N}}^{(\alpha)}[h]} \mathbb{E}|\hat{H}(X_1, \ldots, X_n) - \mathrm{H}(\boldsymbol{\mu})|, \tag{4}$$

where the infimum is over all mappings $\hat{H} : \mathbb{N}^n \to \mathbb{R}_+$.

## 3 Main results

Our first result is a dimension-free analogue of (2):

**Theorem 1.** *For all $\alpha > 1$, $\mathrm{H} : \Delta_{\mathbb{N}}^{(\alpha)} \to \mathbb{R}_+$ is uniformly continuous under $\ell_1$. In particular, for all $\boldsymbol{\mu}, \boldsymbol{\nu} \in \Delta_{\mathbb{N}}^{(\alpha)}$ satisfying $\|\boldsymbol{\mu} - \boldsymbol{\nu}\|_\infty < 1/2$, we have*

$$
\begin{aligned}
|\mathrm{H}(\boldsymbol{\mu}) - \mathrm{H}(\boldsymbol{\nu})| \quad &\leq \quad \|\boldsymbol{\mu} - \boldsymbol{\nu}\|_1^{1-1/\alpha} \left( 2\alpha^\alpha + \mathrm{H}^{(\alpha)}(\boldsymbol{\mu}) + \mathrm{H}^{(\alpha)}(\boldsymbol{\nu}) \right)^{1/\alpha} \\
&\leq \quad \|\boldsymbol{\mu} - \boldsymbol{\nu}\|_1^{1-1/\alpha} \left( 2\alpha + \mathrm{H}^{(\alpha)}(\boldsymbol{\mu})^{1/\alpha} + \mathrm{H}^{(\alpha)}(\boldsymbol{\nu})^{1/\alpha} \right).
\end{aligned}
$$

The requirement in Theorem 1 that $\alpha > 1$ cannot be dispensed with, as the function $\mathrm{H} : \Delta_{\mathbb{N}}^{(\alpha)}[h] \to \mathbb{R}_+$ is not continuous under $\ell_1$ for $\alpha = 1$ (see Remark following Lemma 5), and, a fortiori, is not uniformly continuous. Thus, there can be no function $F : \mathbb{R}_+^2 \to \mathbb{R}_+$ satisfying

$$|\mathrm{H}(\boldsymbol{\mu}) - \mathrm{H}(\boldsymbol{\nu})| \quad \leq \quad F(\|\boldsymbol{\mu} - \boldsymbol{\nu}\|_1, h), \qquad h > 0, \boldsymbol{\mu}, \boldsymbol{\nu} \in \Delta_{\mathbb{N}}^{(1)}[h]$$

with the additional property that for any two sequences $\boldsymbol{\mu}_n, \boldsymbol{\nu}_n \in \Delta_{\mathbb{N}}$ satisfying $\varepsilon_n := \|\boldsymbol{\mu}_n - \boldsymbol{\nu}_n\|_1 \to 0$, it holds that $F(\varepsilon_n, h) \to 0$.

Perhaps surprisingly,[4] it turns out that $\mathrm{H} : \Delta_{\mathbb{N}}^{(\alpha)}[h] \to \mathbb{R}_+$ *is* uniformly continuous under $\ell_p$ for all $\alpha > 1$, $p \in [1, \infty]$:

**Theorem 2.** *There is a function $F : \mathbb{R}_+^4 \to \mathbb{R}_+$ such that*

$$|\mathrm{H}(\boldsymbol{\mu}) - \mathrm{H}(\boldsymbol{\nu})| \quad \leq \quad F(\|\boldsymbol{\mu} - \boldsymbol{\nu}\|_p, h, \alpha, p), \qquad h > 0, \alpha > 1, p \in [1, \infty], \boldsymbol{\mu}, \boldsymbol{\nu} \in \Delta_{\mathbb{N}}^{(\alpha)}[h]$$

*with the additional property that whenever $\varepsilon_n := \|\boldsymbol{\mu}_n - \boldsymbol{\nu}_n\|_p \to 0$, we have $F(\varepsilon_n, h, \alpha, p) \to 0$.*

**Remark.** Although Theorem 2 establishes uniform continuity, it gives no hint as to the functional dependence of the modulus of continuity $F$ on $\alpha$, $p$, $h$, and $\|\boldsymbol{\mu} - \boldsymbol{\nu}\|_p$. We leave this as a fascinating open problem — even though the practical applications are likely to be limited: it follows from Wyner and Foster [2003] and Theorem 4 that for $p = \alpha = 2$ and fixed $h$, $F(\|\boldsymbol{\mu} - \boldsymbol{\nu}\|_2, h, 2, 2)$ cannot decay at a faster rate than $1/\log(1/\|\boldsymbol{\mu} - \boldsymbol{\nu}\|_2)$.

Combining Theorem 1 with (1) yields an empirical (under moment assumptions) bound for the plug-in entropy estimator:

---

[4]Since $\ell_1$ dominates all of the $\ell_p$ norms, continuity of a function under $\ell_p$ trivially implies continuity under $\ell_1$, but the reverse implication is generally not true.

**Corollary 1.** *For all $\alpha > 1$, $h > 0$, $\delta \in (0,1)$, $n \geq 2\log\frac{4}{\delta}$, and $\boldsymbol{\mu} \in \Delta_{\mathbb{N}}^{(\alpha)}[h]$, we have that*

$$|\mathrm{H}(\boldsymbol{\mu}) - \mathrm{H}(\hat{\boldsymbol{\mu}}_n)| \leq \left(2\alpha^\alpha + h + \mathrm{H}^{(\alpha)}(\hat{\boldsymbol{\mu}}_n)\right)^{1/\alpha} \left(\frac{2\|\hat{\boldsymbol{\mu}}_n\|_{1/2}^{1/2}}{\sqrt{n}} + 6\sqrt{\frac{\log(4/\delta)}{2n}}\right)^{1-1/\alpha}$$

*holds with probability at least $1 - \delta$.*

In Section 6, we compare the rates implied by Corollary 1 to the state of the art on various distributions.

Next, we examine the optimality of the plug-in estimate by analyzing the minimax risk, defined in (4). It was known [Silva, 2018, Appendix A] that assuming $\mathrm{H}(\boldsymbol{\mu}) < \infty$ does not suffice to yield a minimax rate for the $L_2$ risk:

$$\inf_{\hat{H}:\mathbb{N}^n \to \mathbb{R}_+} \sup_{\boldsymbol{\mu} \in \Delta_{\mathbb{N}}^{(1)}} \mathbb{E}\left(\hat{H}(X_1, \ldots, X_n) - \mathrm{H}(\boldsymbol{\mu})\right)^2 = \infty.$$

This technique yields an analogous result for the $L_1$ risk as well. We strengthen these results in two ways: (i) by lower-bounding the $L_1$ risk (rather than $L_2$, which is never smaller), and (ii) by restricting $\boldsymbol{\mu}$ to $\Delta_{\mathbb{N}}^{(1)}[h]$ and obtaining a finitary, quantitative lower bound:

**Theorem 3.** *For $\alpha = 1$, there is a universal constant $C > 0$ such that for all $h > 1$ and $n \in \mathbb{N}$, we have $\mathcal{R}_n^{(1)}(h) \geq Ch$.*

**Remark.** The above result complements — but is not directly comparable to — Antos and Kontoyiannis [2001a, Theorem 4]. Ours gives a quantitative dependence on $h$ but constructs an adversarial distribution for each sample size $n$; theirs is asymptotic only but a single adversarial distribution suffices for all $n$.

**Remark.** Our technique immediately yields a lower bound of $Ch^2$ on the $L_2$ minimax risk.

In contradistinction to the $\alpha = 1$ case, where no minimax rate exists, we show that the plug-in estimator is minimax for all $\alpha > 1$:

**Theorem 4.** *The following bounds hold for the $L_1$ minimax risk:*

*(a) Upper bound: for all $h > 0, \alpha > 1$,*

$$\mathcal{R}_n^{(\alpha)}(h) \leq \frac{1 + \log n}{\sqrt{n}} + \frac{2^{\alpha-1}h}{\log^{\alpha-1} n}, \qquad n \in \mathbb{N};$$

*further, this bound is achieved by the plug-in estimate $\mathrm{H}(\hat{\boldsymbol{\mu}}_n)$.*

*(b) Lower bound: for each $\alpha > 0$, $n \in \mathbb{N}$ there is an $h > 0$ such that*

$$\mathcal{R}_n^{(\alpha)}(h) \geq \frac{h}{4 \cdot 3^\alpha \log^{\alpha-1} n}.$$

**Open problem.** Close the gap in the dependence on $\alpha$ in the upper and lower bounds.

## 4 Related work

**Continuity, convergence, moments of information.** Zhang [2007] gave a sharpened version of (2) and Ho and Yeung [2010] presented analogous bounds; Audenaert [2007] proved a non-commutative generalization. Sason [2013, Theorem 5] upper-bounds $|\mathrm{H}(\boldsymbol{\mu}) - \mathrm{H}(\boldsymbol{\nu})|$ in terms of quantities related to $\|\boldsymbol{\mu} - \boldsymbol{\nu}\|_1$, where (at most) one of them is allowed to have infinite support. Even though $\mathrm{H}(\cdot)$ is not continuous on $\Delta_{\mathbb{N}}$, the plug-in estimate $\mathrm{H}(\hat{\boldsymbol{\mu}}_n)$ converges to $\mathrm{H}(\boldsymbol{\mu})$ almost surely and in $L_2$ [Antos and Kontoyiannis, 2001a]. Silva [2018] studied a variety of restrictions on distributions over infinite alphabets to derive strong consistency results and rates of convergence. Moments of information were apparently first defined in Golomb [1966].

**Entropy estimation.**    Recent surveys of entropy estimation results may be found in Jiao et al. [2015], Verdú [2019]. The finite-alphabet case is particularly well-understood. For fixed alphabet size $d < \infty$, the plug-in estimate is asymptotically minimax optimal [Paninski, 2003]. Paninski [2004] non-constructively established the existence of a sublinear (in $d$) entropy estimator. The optimal dependence on $d$ (at fixed accuracy) was settled by Valiant and Valiant [2011a, 2017] as being $\Theta(d/\log d)$.

The $\Theta(d/\log d)$ dependence on the alphabet size is also relevant in the so-called *high dimensional* asymptotic regime, where $d$ grows with $n$. Here, the plug-in estimate is no longer optimal, and more sophisticated techniques are called for [Valiant and Valiant, 2011a,b, 2017]. The works of Wu and Yang [2016], Jiao et al. [2015], Han et al. [2015], Jiao et al. [2017] characterized the minimax rates for the high-dimensional regime: a small additive error of $\varepsilon$ requires $\Theta(d/\varepsilon \log d)$ samples. Building off of these polynomial-approximation based constructions, Acharya et al. [2017] design an additional optimal estimator, this one based on a profile maximum likelihood approach that can also estimate a variety of other important statistics. Fukuchi and Sakuma [2017, 2018] generalize the optimal estimators to estimate any additive functional, recovering in particular the optimal rates for entropy. Acharya et al. [2019] modify these optimal estimators with the added goal of low space complexity.

Finally, there is the infinite-alphabet case. Although here the plug-in estimate is again universally strongly consistent, control of the convergence rate requires some assumption on the sampling distribution — and Antos and Kontoyiannis [2001a] compellingly argue that moment assumptions are natural and minimalistic. Absent any prior assumptions, the $L_1$ (and hence $L_2$) convergence rate of *any* estimator can be made arbitrarily slow (Theorem 4 ibid.). The present paper proves a variant of this result (see Theorem 3 and the Remark following it). Antos and Kontoyiannis [2001a] further show that even under moment assumptions, there is no polynomial rate of convergence for the plug-in estimate: there is no $\beta > 0$ such that its risk decays as $O(n^{-\beta})$. Wyner and Foster [2003] showed that the plug-in estimate achieves a rate of $O(\frac{1}{\log n})$ for bounded second moment, and this is minimax optimal. Brautbar and Samorodnitsky [2007] exhibited a function of the higher moments that can be used in place of alphabet size to give a multiplicative approximation to the entropy.

# 5   Proofs

## 5.1   Proof of Theorem 1

We begin with a subadditivity result for the $\alpha$th moment of information (which we state for $\alpha > 0$, even though only the range $\alpha > 1$ will be needed).

**Lemma 1.** *For $\alpha > 0$ and $\boldsymbol{\mu}, \boldsymbol{\nu} \in \Delta_{\mathbb{N}}^{(\alpha)}$, we have*

$$\mathrm{H}^{(\alpha)}(|\boldsymbol{\mu} - \boldsymbol{\nu}|) \leq 2\alpha^{\alpha} + \mathrm{H}^{(\alpha)}(\boldsymbol{\mu}) + \mathrm{H}^{(\alpha)}(\boldsymbol{\nu}).$$

*Proof.* Define $\mathrm{h}^{(\alpha)}\colon [0,1] \to \mathbb{R}_+$ by $z \mapsto z \ln^{\alpha}(1/z)$, where $\mathrm{h}^{(\alpha)}(0) = 0$. The function $\mathrm{h}^{(\alpha)}$ is increasing on $[0, \mathrm{e}^{-\alpha}]$ and decreasing on $[\mathrm{e}^{-\alpha}, 1]$. The maximum is therefore achieved at $z = \mathrm{e}^{-\alpha}$, and

$$\max_{z \in [0,1]} \mathrm{h}^{(\alpha)}(z) = \mathrm{h}^{(\alpha)}(\mathrm{e}^{-\alpha}) = \mathrm{e}^{-\alpha}\alpha^{\alpha}. \tag{5}$$

Now decompose $\mathrm{H}^{(\alpha)}$:

$$\mathrm{H}^{(\alpha)}(|\boldsymbol{\mu} - \boldsymbol{\nu}|) \;\; = \;\; \sum_{i: \boldsymbol{\mu}(i) \vee \boldsymbol{\nu}(i) > \mathrm{e}^{-\alpha}} \mathrm{h}^{(\alpha)}(|\boldsymbol{\mu}(i) - \boldsymbol{\nu}(i)|) + \sum_{i: \boldsymbol{\mu}(i) \vee \boldsymbol{\nu}(i) \leq \mathrm{e}^{-\alpha}} \mathrm{h}^{(\alpha)}(|\boldsymbol{\mu}(i) - \boldsymbol{\nu}(i)|).$$

For the first term, since $\boldsymbol{\mu} \in \Delta_{\mathbb{N}}$, it must be that $|\{i \in \mathbb{N}\colon \boldsymbol{\mu}(i) > \mathrm{e}^{-\alpha}\}| \leq \mathrm{e}^{\alpha}$, and similarly for $\boldsymbol{\nu}$. Thus,

$$\sum_{i: \boldsymbol{\mu}(i) \vee \boldsymbol{\nu}(i) > \mathrm{e}^{-\alpha}} \mathrm{h}^{(\alpha)}(|\boldsymbol{\mu}(i) - \boldsymbol{\nu}(i)|) \;\; \leq \;\; \left( \left|\{i\colon \boldsymbol{\mu}(i) > \mathrm{e}^{-\alpha}\}\right| + \left|\{i\colon \boldsymbol{\nu}(i) > \mathrm{e}^{-\alpha}\}\right| \right) \max_{z \in [0,1]} \mathrm{h}^{(\alpha)}(z)$$

$$\leq \;\; 2\mathrm{e}^{\alpha}\mathrm{e}^{-\alpha}\alpha^{\alpha} = 2\alpha^{\alpha}.$$

For the second term, notice that when $\boldsymbol{\mu}(i) \vee \boldsymbol{\nu}(i) \leq \mathrm{e}^{-\alpha}$, the monotonicity of $\mathrm{h}^{(\alpha)}$ implies

$$\mathrm{h}^{(\alpha)}(|\boldsymbol{\mu}(i) - \boldsymbol{\nu}(i)|) \leq \mathrm{h}^{(\alpha)}(\boldsymbol{\mu}(i) \vee \boldsymbol{\nu}(i)),$$

and hence

$$
\begin{aligned}
\sum_{i \in \mathbb{N}: \boldsymbol{\mu}(i) \vee \boldsymbol{\nu}(i) \leq \mathrm{e}^{-\alpha}} \mathrm{h}^{(\alpha)}(|\boldsymbol{\mu}(i) - \boldsymbol{\nu}(i)|) \quad &\leq \quad \sum_{i: \in \boldsymbol{\mu}(i) \vee \boldsymbol{\nu}(i) \leq \mathrm{e}^{-\alpha}} \mathrm{h}^{(\alpha)}(\boldsymbol{\mu}(i) \vee \boldsymbol{\nu}(i)) \\
&\leq \quad \sum_{i: \in \boldsymbol{\mu}(i) \vee \boldsymbol{\nu}(i) \leq \mathrm{e}^{-\alpha}} \mathrm{h}^{(\alpha)}(\boldsymbol{\mu}(i)) + \mathrm{h}^{(\alpha)}(\boldsymbol{\nu}(i)) \\
&\leq \quad \mathrm{H}^{(\alpha)}(\boldsymbol{\mu}) + \mathrm{H}^{(\alpha)}(\boldsymbol{\nu}).
\end{aligned}
$$

$\square$

*Proof of Theorem 1.* The concavity argument in the proof of Cover and Thomas [2006, Theorem 17.3.3], immediately implies

$$|\mathrm{H}(\boldsymbol{\mu}) - \mathrm{H}(\boldsymbol{\nu})| \quad \leq \quad \mathrm{H}(|\boldsymbol{\mu} - \boldsymbol{\nu}|).$$

Then, via an application of Hölder's inequality,

$$
\begin{aligned}
\mathrm{H}(|\boldsymbol{\mu} - \boldsymbol{\nu}|) \quad &= \quad \sum_{i \in \mathbb{N}} |\boldsymbol{\mu}(i) - \boldsymbol{\nu}(i)| \log \frac{1}{|\boldsymbol{\mu}(i) - \boldsymbol{\nu}(i)|} \\
&= \quad \sum_{i \in \mathbb{N}} |\boldsymbol{\mu}(i) - \boldsymbol{\nu}(i)|^{1-1/\alpha} \cdot |\boldsymbol{\mu}(i) - \boldsymbol{\nu}(i)|^{1/\alpha} \log \frac{1}{|\boldsymbol{\mu}(i) - \boldsymbol{\nu}(i)|} \\
&\leq \quad \left( \sum_{i \in \mathbb{N}} \left( |\boldsymbol{\mu}(i) - \boldsymbol{\nu}(i)|^{1-1/\alpha} \right)^{1/(1-1/\alpha)} \right)^{1-1/\alpha} \left( \sum_{i \in \mathbb{N}} \left( |\boldsymbol{\mu}(i) - \boldsymbol{\nu}(i)|^{1/\alpha} \log \frac{1}{|\boldsymbol{\mu}(i) - \boldsymbol{\nu}(i)|} \right)^{\alpha} \right)^{1/\alpha} \\
&= \quad \|\boldsymbol{\mu} - \boldsymbol{\nu}\|_1^{1-1/\alpha} \mathrm{H}^{(\alpha)}(|\boldsymbol{\mu} - \boldsymbol{\nu}|)^{1/\alpha}.
\end{aligned}
$$

The claim follows by invoking Lemma 1 and the subadditivity of $t \mapsto t^{1/\alpha}$ for $t \geq 0$ and $\alpha > 1$. $\square$

## 5.2 Proof of Corollary 1

For $h > 1$ and $n \in \mathbb{N}$, put $a_n = (1 - 1/(2n)) \ln(1 - 1/(2n))$ and define the support size $S = S(h, n)$ by $S = \lfloor (1/2n) \exp(2n(h + a_n)) \rfloor$. Consider the distributions $\boldsymbol{\mu}_0 = (1, 0, 0, \dots)$ and $\boldsymbol{\mu}_n$ defined by $\boldsymbol{\mu}_n(1) = 1 - 1/(2n)$, and

$$\boldsymbol{\mu}_n(i) = \frac{1}{2nS}, \qquad 2 \leq i \leq 1 + S(h, n).$$

We compute the Kullback-Leibler divergence and entropy:

$$
\begin{aligned}
D_{\mathrm{KL}}(\boldsymbol{\mu}_0 \| \boldsymbol{\mu}_n) \quad &= \quad \log \frac{1}{1 - 1/(2n)} \leq \frac{1}{1 - 1/(2n)} - 1 \leq \frac{1}{n} \qquad (6) \\
\mathrm{H}(\boldsymbol{\mu}_0) \quad &= \quad 0 \leq h.
\end{aligned}
$$

For $x \geq 2$, always $\lfloor x \rfloor \geq x/2$. Additionally, from $2na_n \geq -1$, and $\frac{1}{2n} \exp(2nh - 1) > 2$, we obtain that $S > (1/4n) \exp(2n(h + a_n))$, hence we also have that $h \geq \mathrm{H}(\boldsymbol{\mu}_n) > h - \frac{1}{2n} \ln 2$. Since $\frac{1}{2x} \ln 2 \leq 1/2$ on $(0, \infty)$ and $h > 1$, it follows that $\mathrm{H}(\boldsymbol{\mu}_n) \geq \frac{h}{2}$, whence $|\mathrm{H}(\boldsymbol{\mu}_0) - \mathrm{H}(\boldsymbol{\mu}_n)| \geq h/2$. To bound the $L_1$ minimax risk (defined in (4)), we invoke Markov's inequality:

$$\mathbb{E}|\hat{H}(X_1, \dots, X_n) - \mathrm{H}(\boldsymbol{\mu})| \quad \geq \quad \frac{h}{4} \mathbb{P}\left( |\hat{H}(X_1, \dots, X_n) - \mathrm{H}(\boldsymbol{\mu})| > \frac{h}{4} \right).$$

It follows via Le Cam's two point method [Tsybakov, 2008, Section 2.4.2] that

$$\mathcal{R}_n^{(1)}(h) \geq \frac{h}{4} \mathrm{e}^{-nD_{\mathrm{KL}}(\boldsymbol{\mu}_0 \| \boldsymbol{\mu})} \geq \frac{h}{4\mathrm{e}},$$

where the second inequality stems from (6).

$\square$

## 5.3 Proof of Theorem 4

We begin with an auxiliary lemma, of possible independent interest.

**Lemma 2.** *For all $\boldsymbol{\mu} \in \Delta_{\mathbb{N}}$ and $n \in \mathbb{N}$, we have*

$$
\mathrm{H}(\boldsymbol{\mu}) \;\geq\; \mathbb{E}\mathrm{H}(\hat{\boldsymbol{\mu}}_n) \;\geq\; \mathrm{H}(\boldsymbol{\mu}) - \inf_{0 < \varepsilon < 1} \left[ \sum_{i \in \mathbb{N} : \boldsymbol{\mu}(i) < \varepsilon} \boldsymbol{\mu}(i) \log \frac{1}{\boldsymbol{\mu}(i)} + \log \left( 1 + \frac{1}{\varepsilon n} \right) \right].
$$

*Proof.* The first inequality follows from Jensen's, since $\mathrm{H}(\cdot)$ is concave and $\mathbb{E}\hat{\boldsymbol{\mu}}_n = \boldsymbol{\mu}$. To prove the second inequality, choose $\varepsilon > 0$, put $J := \{ i \in \mathbb{N} : \boldsymbol{\mu}(i) < \varepsilon \}$, and compute

$$
\begin{aligned}
\mathbb{E}\mathrm{H}(\hat{\boldsymbol{\mu}}_n) &= \mathbb{E}\left[ \sum_{i \in \mathbb{N} \setminus J} \hat{\boldsymbol{\mu}}_n(i) \log \frac{1}{\hat{\boldsymbol{\mu}}_n(i)} + \sum_{i \in J} \hat{\boldsymbol{\mu}}_n(i) \log \frac{1}{\hat{\boldsymbol{\mu}}_n(i)} \right] \\
&\geq \mathbb{E}\left[ \sum_{i \in \mathbb{N} \setminus J} \hat{\boldsymbol{\mu}}_n(i) \log \frac{1}{\hat{\boldsymbol{\mu}}_n(i)} + \left( \sum_{i \in J} \hat{\boldsymbol{\mu}}_n(i) \right) \log \frac{1}{\sum_{i \in J} \hat{\boldsymbol{\mu}}_n(i)} \right] \\
&=: \mathbb{E}\mathrm{H}(\tilde{\boldsymbol{\mu}}_n),
\end{aligned}
$$

where $\tilde{\boldsymbol{\mu}}_n$ is the "collapsed" version of $\hat{\boldsymbol{\mu}}_n$, where all of the masses in $J$ have been replaced by a single mass equal to their sum, and the inequality holds because conditioning reduces entropy [Cover and Thomas, 2006, Eq.(2.157)]. We observe that $\tilde{\boldsymbol{\mu}}_n$ has support size at most $1 + 1/\varepsilon$ and invoke Paninski [2003, Proposition 1]:

$$
\mathbb{E}\mathrm{H}(\tilde{\boldsymbol{\mu}}_n) \;\geq\; \mathrm{H}(\tilde{\boldsymbol{\mu}}) - \log\left( 1 + \frac{1}{\varepsilon n} \right), \tag{7}
$$

where $\tilde{\boldsymbol{\mu}}$ is the "collapsed" version of $\boldsymbol{\mu}$. Now

$$
\begin{aligned}
\mathrm{H}(\tilde{\boldsymbol{\mu}}) &= \mathrm{H}(\boldsymbol{\mu}) + \left( \sum_{i \in j} \boldsymbol{\mu}(i) \right) \log \frac{1}{\sum_{i \in J} \boldsymbol{\mu}(i)} - \sum_{i \in J} \boldsymbol{\mu}(i) \log \frac{1}{\boldsymbol{\mu}(i)} \\
&\geq \mathrm{H}(\boldsymbol{\mu}) - \sum_{i \in J} \boldsymbol{\mu}(i) \log \frac{1}{\boldsymbol{\mu}(i)},
\end{aligned}
$$

which concludes the proof. $\qquad\square$

The first part of the theorem will follow from the following proposition.

**Proposition 1.** *For $\alpha \geq 1$, $h > 0$, $n \in \mathbb{N}$ and $\boldsymbol{\mu} \in \Delta_{\mathbb{N}}^{(\alpha)}[h]$, we have*

$$
\mathbb{E}|\mathrm{H}(\boldsymbol{\mu}) - \mathrm{H}(\hat{\boldsymbol{\mu}}_n)| \;\leq\; \frac{\log n}{\sqrt{n}} + \inf_{0 < \varepsilon < 1} \left[ \left( \log \frac{1}{\varepsilon} \right)^{1 - \alpha} h + \log \left( 1 + \frac{1}{\varepsilon n} \right) \right].
$$

*Proof.* Since by Lemma 2, $|\mathrm{H}(\boldsymbol{\mu}) - \mathbb{E}\mathrm{H}(\hat{\boldsymbol{\mu}}_n)| = \mathrm{H}(\boldsymbol{\mu}) - \mathbb{E}\mathrm{H}(\hat{\boldsymbol{\mu}}_n)$, it follows from the triangle and Jensen inequalities that

$$
\begin{aligned}
\mathbb{E}|\mathrm{H}(\boldsymbol{\mu}) - \mathrm{H}(\hat{\boldsymbol{\mu}}_n)| &\leq \mathbb{E}|\mathrm{H}(\hat{\boldsymbol{\mu}}_n) - \mathbb{E}\mathrm{H}(\hat{\boldsymbol{\mu}}_n)| + \mathrm{H}(\boldsymbol{\mu}) - \mathbb{E}\mathrm{H}(\hat{\boldsymbol{\mu}}_n) \\
&\leq \sqrt{\mathbb{V}\mathbf{ar}\left[ \mathrm{H}(\hat{\boldsymbol{\mu}}_n) \right]} + \mathrm{H}(\boldsymbol{\mu}) - \mathbb{E}\mathrm{H}(\hat{\boldsymbol{\mu}}_n) \\
&\leq \frac{\log n}{\sqrt{n}} + \mathrm{H}(\boldsymbol{\mu}) - \mathbb{E}\mathrm{H}(\hat{\boldsymbol{\mu}}_n), \tag{8}
\end{aligned}
$$

where the variance bound is from Antos and Kontoyiannis [2001b, Proposition 1(iv)].

For any $\varepsilon > 0$, Lemma 2 implies

$$
\begin{aligned}
\mathbb{E}\mathrm{H}(\hat{\boldsymbol{\mu}}_n) \;\geq\; & \mathrm{H}(\boldsymbol{\mu}) - \sum_{i \in \mathbb{N}:\boldsymbol{\mu}(i)<\varepsilon} \boldsymbol{\mu}(i)\log\frac{1}{\boldsymbol{\mu}(i)} - \log\left(1+\frac{1}{\varepsilon n}\right) \\
\geq\; & \mathrm{H}(\boldsymbol{\mu}) - \left(\log\frac{1}{\varepsilon}\right)^{1-\alpha} \sum_{i \in \mathbb{N}:\boldsymbol{\mu}(i)<\varepsilon} \boldsymbol{\mu}(i)\left(\log\frac{1}{\boldsymbol{\mu}(i)}\right)^{\alpha} - \log\left(1+\frac{1}{\varepsilon n}\right) \\
\geq\; & \mathrm{H}(\boldsymbol{\mu}) - \left(\log\frac{1}{\varepsilon}\right)^{1-\alpha} \mathrm{H}^{(\alpha)}(\boldsymbol{\mu}) - \log\left(1+\frac{1}{\varepsilon n}\right),
\end{aligned}
\tag{9}
$$

where the second and third inequalities follow from the obvious relations

$$
\sum_{i:\boldsymbol{\mu}(i)<\varepsilon} \boldsymbol{\mu}(i)\log\frac{1}{\boldsymbol{\mu}(i)} \;\leq\; \left(\log\frac{1}{\varepsilon}\right)^{1-\alpha} \sum_{i:\boldsymbol{\mu}(i)<\varepsilon} \boldsymbol{\mu}(i)\left(\log\frac{1}{\boldsymbol{\mu}(i)}\right)^{\alpha} \;\leq\; \left(\log\frac{1}{\varepsilon}\right)^{1-\alpha} \mathrm{H}^{(\alpha)}(\boldsymbol{\mu}).
$$

The claim follows by combining (8) with (9). $\qquad\square$

*Proof of Theorem 4(a).* Use the fact that $\mathcal{R}_n^{(\alpha)}(h) \leq \mathbb{E}|\mathrm{H}(\boldsymbol{\mu}) - \mathrm{H}(\hat{\boldsymbol{\mu}}_n)|$, invoke Proposition 1 with $\varepsilon = \frac{1}{\sqrt{n}}$ and use $\log(1+x) \leq x$. $\qquad\square$

We now prove the second half of the theorem.

*Proof of Theorem 4(b).* Let $\alpha > 0$, $n \in \mathbb{N}$ and define two families of distributions:

$$
\mathcal{U}_1 := \left\{\boldsymbol{\mu}_1 = \mathrm{Uniform}([n^3])\right\}, \quad \mathcal{U}_2 := \left\{\boldsymbol{\mu}_2 = \mathrm{Uniform}(A) : A \subset [n^3], |A| = n^2\right\}.
$$

Let $h := 3^{\alpha}\log^{\alpha} n$ and note that $\mathcal{U}_1 \cup \mathcal{U}_2 \subseteq \Delta_{\mathbb{N}}^{(\alpha)}[h]$. Let $E$ be the event that $\boldsymbol{X} = (X_1, \ldots, X_n)$ has no repeating elements, i.e $|\{X_1, X_2, \ldots, X_n\}| = n$. Let $\boldsymbol{\mu}_1 \in \mathcal{U}_1, \boldsymbol{\mu}_2 \in \mathcal{U}_2$ and consider the values $\mathbb{P}_{\boldsymbol{X}\sim\boldsymbol{\mu}_1^n}(E)$ and $\mathbb{P}_{\boldsymbol{X}\sim\boldsymbol{\mu}_2^n}(E)$. For $m \in \mathbb{N}$, define $\mathcal{X}(m)$ to be the smallest $k$ such that when uniformly throwing $m$ balls into $k$ buckets, the probability of collision is at least $1/2$. Since $\mathcal{X}(m)$ is known[5] to be at least $\sqrt{m}$ (and hence $\mathcal{X}(n^2) > n$) we have a lower bound of $\frac{1}{2}$ on both $\mathbb{P}_{\boldsymbol{X}\sim\boldsymbol{\mu}_1^n}(E)$ and $\mathbb{P}_{\boldsymbol{X}\sim\boldsymbol{\mu}_2^n}(E)$. Define $\boldsymbol{\mu}_1^n|E$ as the distribution on $\mathbb{N}^n$ induced by conditioning the product $\boldsymbol{\mu}_1^n$ on the event $E$, and define $\boldsymbol{\mu}_2^n|E$ analogously. Our key observation is that conditional on $E$, (i) both are effectively distributions on ordered $n$-tuples from $[n^3]$, and (ii) $\boldsymbol{\mu}_1^n$ is uniform on $([n^3])_n$ whereas $\boldsymbol{\mu}_2^n = \mathrm{Uniform}(A)$ is uniform on $(A)_n$, where $(J)_k := \left\{(x_1, \ldots, x_k) \in J^k : |\{x_1, \ldots, x_k\}| = k\right\}$, $\qquad J \subset \mathbb{N}, k \in \mathbb{N}$. Then

$$
\begin{aligned}
\mathcal{R}_n^{(\alpha)}(h) &\geq \inf_{\hat{H}} \sup_{\boldsymbol{\mu}\in\mathcal{U}_1\cup\mathcal{U}_2} \mathbb{E}_{\boldsymbol{X}\sim\boldsymbol{\mu}^n}\left[|\hat{H}(\boldsymbol{X}) - \mathrm{H}(\boldsymbol{\mu})|\right] \\
&\overset{(a)}{\geq} \inf_{\hat{H}} \sup_{\boldsymbol{\mu}\in\mathcal{U}_1\cup\mathcal{U}_2} \mathbb{E}_{\boldsymbol{X}\sim\boldsymbol{\mu}^n|E}\left[|\hat{H}(\boldsymbol{X}) - \mathrm{H}(\boldsymbol{\mu})|\right] \mathbb{P}_{\boldsymbol{X}\sim\boldsymbol{\mu}^n}(E) \\
&\geq \inf_{\hat{H}} \frac{1}{2} \sup_{\boldsymbol{\mu}\in\mathcal{U}_1\cup\mathcal{U}_2} \mathbb{E}_{\boldsymbol{X}\sim\boldsymbol{\mu}^n|E}\left[|\hat{H}(\boldsymbol{X}) - \mathrm{H}(\boldsymbol{\mu})|\right] \\
&\overset{(b)}{\geq} \inf_{\hat{H}} \frac{1}{4} \left(\mathbb{E}_{\boldsymbol{X}\sim\boldsymbol{\mu}_1^n|E}\left[|\hat{H}(\boldsymbol{X}) - \mathrm{H}(\boldsymbol{\mu}_1)|\right] + \sup_{\boldsymbol{\mu}_2\in\mathcal{U}_2} \mathbb{E}_{\boldsymbol{X}\sim\boldsymbol{\mu}_2^n|E}\left[|\hat{H}(\boldsymbol{X}) - \mathrm{H}(\boldsymbol{\mu}_2)|\right]\right) \\
&\overset{(c)}{\geq} \inf_{\hat{H}} \frac{1}{4} \left(\mathbb{E}_{\boldsymbol{X}\sim\boldsymbol{\mu}_1^n|E}\left[|\hat{H}(\boldsymbol{X}) - \mathrm{H}(\boldsymbol{\mu}_1)|\right] + \mathbb{E}_{\boldsymbol{\mu}_2\sim\mathrm{Uniform}(\mathcal{U}_2)}\left[\mathbb{E}_{\boldsymbol{X}\sim\boldsymbol{\mu}_2^n|E}\left[|\hat{H}(\boldsymbol{X}) - \mathrm{H}(\boldsymbol{\mu}_2)|\right]\right]\right) \\
&\overset{(d)}{=} \inf_{\hat{H}} \frac{1}{4} \left(\mathbb{E}_{\boldsymbol{X}\sim\boldsymbol{\mu}_1^n|E}\left[|\hat{H}(\boldsymbol{X}) - \mathrm{H}(\boldsymbol{\mu}_1)|\right] + \mathbb{E}_{\boldsymbol{X}\sim\boldsymbol{\mu}_1^n|E}\left[|\hat{H}(\boldsymbol{X}) - \mathrm{H}(\boldsymbol{\mu}_2)|\right]\right) \\
&= \inf_{\hat{H}} \frac{1}{4} \left(\mathbb{E}_{\boldsymbol{X}\sim\boldsymbol{\mu}_1^n|E}\left[|\hat{H}(\boldsymbol{X}) - \mathrm{H}(\boldsymbol{\mu}_1)| + |\hat{H}(\boldsymbol{X}) - \mathrm{H}(\boldsymbol{\mu}_2)|\right]\right) \\
&\overset{(e)}{\geq} \frac{1}{4}|\mathrm{H}(\boldsymbol{\mu}_1) - \mathrm{H}(\boldsymbol{\mu}_2)| = \frac{1}{4}\log n = \frac{1}{4}\frac{h}{3^{\alpha}\log^{\alpha-1} n},
\end{aligned}
$$

---

[5]Better bounds exist [Brink, 2012].

where (a) is from the law of total expectation (the complement of $E$ is discarded), (b) and (c) are bounding a supremum by an average, (e) is from the triangle inequality, and (d) is by observing that, by symmetry, the operators $\mathbb{E}_{\boldsymbol{\mu}_2 \sim \mathrm{Uniform}(\mathcal{U}_2)} \left[ \mathbb{E}_{\boldsymbol{X} \sim \boldsymbol{\mu}_2^n | E} [\cdot] \right]$ and $\mathbb{E}_{\boldsymbol{X} \sim \boldsymbol{\mu}_1^n | E} [\cdot]$ are equivalent. (There is a minor abuse of notation in transitions after (c), since we write $\boldsymbol{\mu}_2$ without specifying a *particular* member of $\mathcal{U}_2$. However, $\boldsymbol{\mu}_2$ only occurs therein as $\mathrm{H}(\boldsymbol{\mu}_2)$, and this value is identical for all $\boldsymbol{\mu}_2 \in \mathcal{U}_2$.) $\qquad\square$

# 6 Rates

Our bounds have the crucial characteristic of being empirical (under moment assumptions). When we *observe* favorable distributions (even without a priori knowledge of the fact), we will benefit from tighter bounds. This entails some cost, and in the worst case our bounds will be sub-optimal. In this section, we illustrate these trade-offs for various natural classes of distributions.

For the class of all finite alphabet distributions, our bound is sub-optimal. The MLE (plug-in estimator) is competitive with the optimal estimator up to logarithmic factors in $d$, but our bounds on the MLE are loose nearly quadratically in $d/n$, in the worst case. The convergence of the empirical distribution on a finite alphabet in $\ell_1$ occurs at rate $\Theta(\sqrt{d/n})$, whereas the MLE entropy estimator converges at rate $O\left( \sqrt{\left(\frac{d}{n}\right)^2 + \frac{\log^2 d}{n}} \right)$, as follows from Wu and Yang [2016, Proposition 1]. So any approach that upper bounds the entropy risk via $\ell_1$ (as our Theorem 1 or Section 4 of Ho and Yeung [2010]) will be worst-case suboptimal for this class of distributions.

Nevertheless, for certain classes of distributions our bounds (Theorem 1 and Corollary 1) can significantly outperform the state of the art, for small and moderate-sized samples. To calculate the expected rate of our approach, we apply Hölder's inequality, as in the proof of Theorem 1:

$$\mathbb{E}|\mathrm{H}(\hat{\boldsymbol{\mu}}_n) - \mathrm{H}(\boldsymbol{\mu})| \leq \left( \mathbb{E}\left[ 2\alpha^\alpha + \mathrm{H}^{(\alpha)}(\boldsymbol{\mu}) + \mathrm{H}^{(\alpha)}(\hat{\boldsymbol{\mu}}_n) \right] \right)^{1/\alpha} \left( \mathbb{E}\|\hat{\boldsymbol{\mu}}_n - \boldsymbol{\mu}\|_1 \right)^{1-1/\alpha} .$$

Now, as in the proof of Lemma 1 (recall that $\mathrm{h}^{(\alpha)}(z) := z \ln^\alpha(1/z)$),

$$
\begin{aligned}
\mathbb{E}\mathrm{H}^{(\alpha)}(\hat{\boldsymbol{\mu}}_n) &= \sum_{i \in [d]} \mathbb{E}\mathrm{h}^{(\alpha)}(\hat{\boldsymbol{\mu}}_n(i)) \\
&\leq e^{\alpha-1} \max_{z \in [0, e^{1-\alpha}]} \mathrm{h}^{(\alpha)}(z) + \sum_{\substack{i \in [d] \\ \hat{\boldsymbol{\mu}}_n(i) < e^{1-\alpha}}} \mathbb{E}\mathrm{h}^{(\alpha)}(\hat{\boldsymbol{\mu}}_n(i)) \\
&\overset{(i)}{\leq} e^{\alpha-1} \max_{z \in [0, e^{1-\alpha}]} \mathrm{h}^{(\alpha)}(z) + \mathrm{H}^{(\alpha)}(\boldsymbol{\mu}) \overset{(ii)}{\leq} \frac{\alpha^\alpha}{e} + \mathrm{H}^{(\alpha)}(\boldsymbol{\mu}),
\end{aligned}
$$

where $(i)$ follows from Jensen's inequality and $(ii)$ from (5).

By Berend and Kontorovich [2013, Lemma 6], we have $\mathbb{E}\|\hat{\boldsymbol{\mu}}_n - \boldsymbol{\mu}\|_1 \leq \Lambda_n(\boldsymbol{\mu})$, where

$$\Lambda_n(\boldsymbol{\mu}) := 2 \sum_{\boldsymbol{\mu}(j) < 1/n} \boldsymbol{\mu}(j) + \frac{1}{\sqrt{n}} \sum_{\boldsymbol{\mu}(j) \geq 1/n} \sqrt{\boldsymbol{\mu}(j)}.$$

This quantity is always finite and $\Lambda_n(\boldsymbol{\mu}) \underset{n \to \infty}{\longrightarrow} 0$ for all $\boldsymbol{\mu} \in \Delta_{\mathbb{N}}$ (ibid). Thus, we obtain the bound

$$\mathbb{E}|\mathrm{H}(\hat{\boldsymbol{\mu}}_n) - \mathrm{H}(\boldsymbol{\mu})| \leq \left( \frac{\alpha^\alpha}{e} + 2\alpha^\alpha + 2\mathrm{H}^{(\alpha)}(\boldsymbol{\mu}) \right)^{1/\alpha} \Lambda_n(\boldsymbol{\mu})^{1-1/\alpha}. \tag{10}$$

**Finite support.** For distributions with a large support but concentrated mass, the bound in (10) compares favorably to the state of the art, especially for smaller sample sizes. To illustrate this, consider a mixture of two distributions with support sizes $d$ and $D$: $\boldsymbol{\mu}'$ is uniform over $[d]$, $\boldsymbol{\mu}''$ is uniform over $d + [D]$, and $\boldsymbol{\mu} := p\boldsymbol{\mu}' + (1-p)\boldsymbol{\mu}''$, for some $p \in [0, 1]$.

The state-of-the-art upper bound for the plug-in estimator can be inferred from Wu and Yang [2016, Appendix D], and has the form

$$\mathbb{E}|\mathrm{H}(\hat{\boldsymbol{\mu}}_n) - \mathrm{H}(\boldsymbol{\mu})| \leq \mathrm{WY}(d, D, p, n) := \frac{d + D}{n} + \min\left( C\frac{\log(d+D)}{\sqrt{n}}, \frac{\log n}{\sqrt{n}} \right)$$

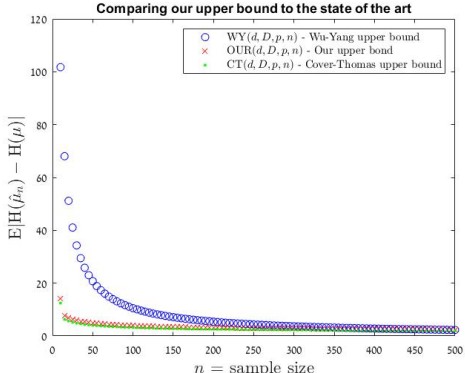
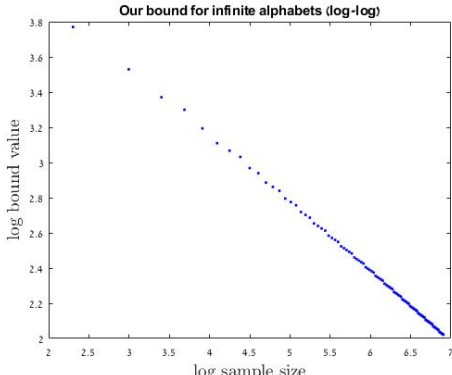

Figure 1: Left: A comparison of the three bounds for $d = 10$, $D = 1000$, $p = 0.95$. Our bound considerably outperforms Wu and Yang [2016] on small samples, and performs nearly as well as the finite-dimensional Cover-Thomas bound. Right: for our value of $q = 2$, the log-log plot shows roughly the correct slope of $-1/2$.

for some $C > 1$; notice that it is insensitive to $p$. For a fair comparison to (10), our estimator's only a priori knowledge of $\boldsymbol{\mu}$ is that its support is of size at most $d + D$. By Proposition 2, we have $\max_{\boldsymbol{\mu} \in \Delta_K} \mathrm{H}^{(\alpha)}(\boldsymbol{\mu}) \leq \max\{\alpha, \log K\}^\alpha + (\alpha/e)^\alpha$. This allows us to optimize over $\alpha$ for each $n$:

$$\mathrm{OUR}(d, D, p, n) := \inf_{\alpha > 1} \left( \frac{\alpha^\alpha}{e} + 2\alpha^\alpha + 2\max\{\alpha, \log(d+D)\}^\alpha + 2(\alpha/e)^\alpha \right)^{1/\alpha} \Lambda_n(\boldsymbol{\mu})^{1-1/\alpha}.$$

Since $\boldsymbol{\mu}$ has finite support, the Cover-Thomas inequality (2) also applies to yield an adaptive estimate when combined with (1). As $t \log(1/t)$ is concave, the latter has the form

$$\mathbb{E}|\mathrm{H}(\hat{\boldsymbol{\mu}}_n) - \mathrm{H}(\boldsymbol{\mu})| \leq \mathbb{E}\left[ \|\hat{\boldsymbol{\mu}}_n - \boldsymbol{\mu}\|_1 \log \frac{d+D}{\|\hat{\boldsymbol{\mu}}_n - \boldsymbol{\mu}\|_1} \right] \leq \Lambda_n(\boldsymbol{\mu}) \log \frac{d+D}{\Lambda_n(\boldsymbol{\mu})} =: \mathrm{CT}(d, D, p, n).$$

The comparisons are plotted in Figure 1 (Left).

**Infinite support.** In some cases our bound is nearly tight (at least for the plug-in estimate), such as for the family of zeta distributions $\boldsymbol{\mu}_q(i) \sim 1/i^q$ with parameter $q > 1$. For this family, Antos and Kontoyiannis [2001a, Theorem 7] establish a lower bound of order $n^{\frac{1-q}{q}}$ on $\mathbb{E}|\mathrm{H}(\hat{\boldsymbol{\mu}}_n) - \mathrm{H}(\boldsymbol{\mu}_q)|$. It is straightforward to verify[6] that $\boldsymbol{\mu}_q \in \Delta_{\mathbb{N}}^{(\alpha)}$ for all $q, \alpha > 1$. Thus, we can optimize our bound in (10) over all $\alpha > 1$; the results are presented in Figure 1 (Right).

## Acknowledgments and Disclosure of Funding

This research was partially supported by the Israel Science Foundation (grant No. 1602/19) and Amazon Research Award. We thank Ioannis Kontoyiannis, Igal Sason, and Sergio Verdú for enlightening conversations.

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
