# OpenReview forum: "Dimension-free empirical entropy estimation"
_NeurIPS.cc/2021/Conference — NeurIPS 2021 Poster_

### Official Review · Reviewer_BrT9 · 2021-07-16

**Rating:** 7
**Confidence:** 4

**Summary:**

Entropy estimation is one of the well studied and fundamental problems in the field of Information theory. The main goal of the paper is to study the entropy estimation by replacing the finite-support assumption (earlier works) with moment conditions. Under these moment conditions, the paper presents various interesting results (Theorem 1-4).

**Limitations And Societal Impact:**

I would urge authors to think about both the potential societal impact and the practical limitations of their work.

**Main Review:**

The things that I like about the paper:
1) I think the question considered in this paper is a natural generalization of the existing research. It is a very well motivated question from a theoretical perspective.
2) The results presented in this paper are quite interesting and almost complete (except for the open problem posed by the authors).
3)The paper is well structured and does a clear presentation of the results. The authors do a good job in placing their contribution with respect to the earlier work.
4) This paper does both theoretical understanding and simulations.

Few concerns/suggestions:
1) As this is the first paper to study the entropy estimation problem under the moment conditions, I would love to see some practical applications. Do authors have any application in mind where the previous estimators were not sufficient?
2) I would encourage authors to include an overview of techniques section where they describe the techniques that were used to prove the theorems and also comment about their novelty/difficulty.

**Time Spent Reviewing:**

8

---

> ### Author Response · Authors · 2021-08-10
> **Thank you for the encouraging feedback.**
>
> **Comment:** *As this is the first paper to study the entropy estimation problem under the moment conditions, I would love to see some practical applications. Do authors have any application in mind where the previous estimators were not sufficient?*
>
> We did not dwell on applications since entropy estimation is such a
> well-studied problem, with dozens of papers listing all sorts of
> motivation and applications. As just one example (among many!), the
> paper of
> Cai, Kulkarni, Verdú. Universal entropy estimation via block sorting.
> IEEE Trans. Inf. Theory 50(7): 1551-1561, 2004
> mentions applications of entropy estimation in neuroscience,
> high-dimensional sources, and data compression, with references given
> therein. In all of these cases, allowing unbounded alphabet size
> significantly expands the scope of applicability.
>  For example, our approach is amenable to any parametric family of distributions (such as the zeta distributions discussed in the paper) where the support size is not bounded but the entropy-moments can be calculated.
> The previous estimates could not handle such distribution families.
>
> We will include a brief survey of such applications in the revision.
>
>
> **Comment:** *I would encourage authors to include an overview of techniques section where they describe the techniques that were used to prove the theorems and also comment about their novelty/difficulty.*
>
> Thank you for the suggestion, we will incorporate it in the revision.
>
> **Comment:** *I would urge authors to think about both the potential societal impact and the practical limitations of their work.*
>
> We will spell out more clearly the limitations of the moment assumption, but we still think that the impact on society at large will be negligible.

---

> > ### Comment · Reviewer_BrT9 · 2021-09-01
> > **Maintain my score**
> >
> > I had minor concerns and the authors have reasonably addressed these concerns.
> >
> > I also read the feedback from the other reviewers and am leaning towards accepting the paper. Therefore, I would like to maintain my score. Thanks!

---

### Official Review · Reviewer_5ypC · 2021-07-16

**Rating:** 7
**Confidence:** 3

**Summary:**

This paper proves an inequality for estimating the entropy by means of plugging in the empirical distribution that depends only empirical quantities, in the setting of iid samples from a discrete alphabet with potentially infinite size. This enables the statistician to certify directly in terms of the sample how well the entropy is estimated by the simple plug-in estimator. The result is proven under a mild moment condition which is known to be essential for consistent estimation, and further the authors show strong lower bounds on the risk when the moment assumption fails to hold. Further, minimax lower bounds are shown for estimating the entropy under this moment assumption that are sharp up to constant factors. In the final section these bounds are compared to pre-existing ones in various settings.


**Limitations And Societal Impact:**

Minor issues/typos:
- p1 Lines 31-34. I believe that the discussion in terms of the alphabet size $d$ is mixed up. I thought Paninski [2003] proved that for fixed $d$ that as the sample size $n$ grows large that the plug-in estimator is optimal? And Wu-Yang [2016] considered the case where $d$ is allowed to grow with $n$ using estimators based on polynomial approximates?
- p4 Line 123: typo: "was was"
- p8 Line 212: Is the MLE the same as the plug-in estimator? If so, I would just stick with the "plug-in" terminology; otherwise please define the MLE.
- p9 Line 216: I think that $O\left( \sqrt{ \frac{d^2}{n} + \frac{\log^2 d}{\sqrt{n}} } \right)$ should be replaced with $O\left( \sqrt{ \frac{d^2}{n} + \frac{\log^2 d}{n} } \right)$
- p9 Line 237: replace "convex" with "concave"

**Main Review:**

**Originality**: This family of results is new to my knowledge. The technique leverages recent empirical bounds on estimating discrete distributions in total variation from Cohen et al [2020] as well as a dimension-free (infinite alphabet) version of a classic inequality of Cover and Thomas. It seems clear that the infinite-alphabet case has not yet been studied from this point of view, and the surrounding literature is well-summarized. Some new interesting open problems are raised as well (understanding the modulus of continuity for Theorem 2 as well as sharpening the constant in the minimax rates of Theorem 4).

**Quality**: Overall the quality of the theoretical results is strong. I have not read the proofs carefully, but the arguments appear to be quite clean and mostly fit inside of the main text. It should be stated in the main text that Theorem 2 is proven in the Appendix. The authors acknowledge that in some cases their bounds are not tight and provide evidence as such in Section 6. Indeed, one expects to lose something in some regimes given the empirical/non-asymptotic nature of these bounds. I found the synthetic evaluation helpful in terms of demonstrating the strength of the bounds derived for small sample sizes as well as applying them to some infinite-alphabet scenarios. It is slightly disappointing though that the derived bounds don't outperform standard Cover-Thomas in Figure 1 Left. It would also be interesting to graph the empirical bound in the zeta distribution example Figure 1 Right.

**Clarity**: The paper is carefully written and well-organized for the most part. All assumptions and theoretical results are clearly stated. My only suggestion is to state explicitly that in Section 6 you are deriving upper bounds that will be compared in Theorem 1. I got a bit confused reading it because the figure is on the previous page. The legend in Figure 1 Left could be improved by including the notation introduced in Section 6 for the various bounds.

**Significance**: This paper provides some state-of-the-art bounds for entropy estimation from a few perspectives in the infinite alphabet case: empirical bounds, minimax rates under moment conditions, and a dimension-free variant of the Cover-Thomas bound. The open problems may attract attention as well.

**Time Spent Reviewing:**

7

---

> ### Author Response · Authors · 2021-08-10
> **Thank you for the encouraging feedback.**
>
> **Comment:** *My only suggestion is to state explicitly that in Section 6 you are deriving upper bounds that will be compared in Theorem 1*
>
> We will incorporate this suggestion into the revision.
>
> **Comment:** *It would also be interesting to graph the empirical bound in the zeta distribution example Figure 1 Right.*
>
> Great suggestion, we will add this.
>
> **Comment:** *The legend in Figure 1 Left could be improved by including the notation introduced in Section 6 for the various bounds.*
>
> We will incorporate this suggestion into the revision.
>
> **Comment:** *p1 Lines 31-34. I believe that the discussion in terms of the alphabet size  is mixed up. I thought Paninski [2003] proved that for fixed  that as the sample size  grows large that the plug-in estimator is optimal? And Wu-Yang [2016] considered the case where $d$ is allowed to grow with $n$ using estimators based on polynomial approximates?*
>
> We were trying to make 2 points: (i) for d<infty, plug-in is not optimal, and (ii) for any fixed d, including d=infty, plug-in is optimal. (See
> Corollary 1 in Antos & Kontonyiannis, last paragraph on p.1 of
> Wu & Yang, and Paninski first paragraph of Sec. 3.1.)
> We will rephrase this paragraph to make it clearer.
>
> **Comment:** *p8 Line 212: Is the MLE the same as the plug-in estimator? If so, I would just stick with the "plug-in" terminology; otherwise please define the MLE.*
>
> Yes, they are the same. Thank you for pointing this out, will clarify.
>
> **Comment:** *It is slightly disappointing though that the derived bounds don't outperform standard Cover-Thomas in Figure 1 Left.*
>
> On the contrary, we argue that one should be pleasantly surprised that our more general bound, which holds for infinite alphabet sizes, did not perform worse than the more restrictive finite-alphabet bound of Cover-Thomas.
>
>
> Thank you for the remaining grammar/stylistic suggestions; we will incorporate them into the revision.

---

> > ### Comment · Reviewer_5ypC · 2021-08-24
> > **Thanks for the reply**
> >
> > Thanks for the comments and addressing my concerns and suggestions.

---

### Official Review · Reviewer_9i5x · 2021-07-17

**Rating:** 7
**Confidence:** 4

**Summary:**

The main contribution of the paper is to give a continuity bound on entropy: how far the entropies of two categorical distributions are, based on the L1-distance between these distributions. This bound does not depend on dimension, and is instead expressed in terms of a higher information-moment, which is similar to the expression of entropy, but with the log taken to a power alpha. Let’s call this alpha-entropy (note: not Renyi entropy).

Using high-probability bounds on the L1-distance between empirical and true distributions, this gives a parallel bound on the accuracy of the empirical plug-in entropy estimator. The paper shows that, over the class of bounded alpha-entropy distributions, the plug-in estimator is minimax optimal. For the class of finite-dimensional distributions, the plug-in estimator is not minimax optimal, but the paper also shows that for sub-classes with small alpha-entropy its rate is not that far-off from the optimal rate.

**Ethical Concerns:**

There are no ethical concerns in this paper.

**Limitations And Societal Impact:**

It's hard to connect this work to any clear limitations or negative societal impact.

**Main Review:**

### Strengths:

+ Extending the Cover-Thomas continuity bound in this way is beautiful, and can suggest other potential extensions. This is worthwhile sharing with the community. _(originality)_

+ The paper, despite being quite technical, is clearly exposed. The proofs, as far as I can see, are correct and well-explained. _(quality, clarity)_

### Weaknesses:

- The paper makes one key claim which is not fully supported. It’s the idea that the bound is completely empirical and can be used to judge the quality of the plug-in estimator entirely from data. But this is only true under the moment assumption, because a non-empirical quantity still remains in the bound (Corollary 1). Even later in the paper this not-entirely-true claim is reiterated (`line 208`). _(quality)_

- A lot of the significance of the result hinges on how meaningful the new moment condition is, what kind of distribution families it defines, and mainly whether it is a good quantifier of the difficulty of estimating entropy. This is not quite elaborated, with the closest argument in the paper, the lower bound which says that for alpha=1 we can’t get a vanishing minimax risk, answering this only in an extreme case. _(significance)_

### Suggestions:

* Theorem 4(a)’s upper bound form is not obvious from the main bounds. Reading through the proofs elucidates it, but I think the transition should be more front and center in the main body of the paper.
* I’m not sure if the term on `line 85` is correct.
* `line 64`, remove one _from_
* Esoteric words like _finitary_ and _contradistinction_ can be distracting

[Edit: Thank you authors for your detailed response. I am raising my evaluation, and I hope to see some of the discussion in your response in the paper itself!]

**Time Spent Reviewing:**

3

---

> ### Author Response · Authors · 2021-08-10
> **Thank you for the encouraging feedback.**
>
> **Comment:** *claim which is not fully supported. It’s the idea that the bound is completely empirical*
>
> Although we do state in the abstract that fully empirical entropy estimation is impossible (and restate it in the introduction, with a proof sketch), we will tone down this claim to make it absolutely clear that we are not achieving completely empirical estimates.
> Thank you for affirming the significance of an empirical bound, even if it is not achievable in full for entropy absent the additional moment assumption.
>
> **Comment:** *A lot of the significance of the result hinges on how meaningful the new moment condition is, what kind of distribution families it defines, and mainly whether it is a good quantifier of the difficulty of estimating entropy.*
>
> We agree that these are excellent questions, very much worthy of follow-up efforts.
> Certainly, the geometric and the zeta distribution families (the latter defined in our paper) are examples that are amenable to our approach but not to the previous approaches that assume finite alphabet size. Our basic
> justification for the moment assumption is that it is *considerably* less restrictive than the finite-alphabet assumption.
> As we argue in lines 35-43, the moment assumption is, in some sense, a minimally restrictive condition under which empirical confidence bounds are possible.
>
> As entropy is the first moment of $\log(1/p(x))$, a natural way around the impossibility-of-estimation result is to assume bounds on higher moments. This relates to the discussion at lines 35-43, and essentially is the desideratum laid out by [Antos and Kontoyiannis 2001a].
> It generalizes both finite distributions and tail conditions studied by previous works. One popular family with infinite support, but small moments is the geometric distributions, as in Example 1.4 of [Brautbar and Samordnitsky 2007]. Finally, an auxiliary constant-factor matching lower bound to Theorem 1 also lends support to quantifying distribution complexity via moments.
>
> In terms of scientific precedent, our moment conditions are very much in the spirit of Section 4.1 of [Antos and Kontonyiannis 2001a]. There, the authors hypothesize that -- in parallel to the asymptotic distribution for the finite alphabet case -- moment conditions are the right notion to achieve estimability in the infinite alphabet case. Although, for this reason, moment conditions are their ideally desired notion, in order to successfully prove convergence of the plug-in, they were required to make stronger tail assumptions on the distribution. As they state ([Antos and Kontonyiannis 2001a] p.165), "under the milder condition [of bounded second moment] ... we were unable to provide upper bounds for the convergence of the plug-in estimates." Or, again ([Antos and Kontonyiannis 2001b] p.45), "as it turns out, the intuition one gains from looking at the finite-alphabet case does not go through in the case of infinite alphabets." Absent tail conditions, they can instead only show a negative result (Corr. 5) that there is no algebraic rate of convergence. We, on the other hand (following [Wyner and Foster 2003]), show a positive result, that *there is* a (inverse) logarithmic convergence rate (Thm. 4(a)). Furthermore, using empirical quantities, this rate can be very much accelerated (Thm. 1).
>
> **Comment:** *Theorem 4(a)’s upper bound form is not obvious from the main bounds. Reading through the proofs elucidates it, but I think the transition should be more front and center in the main body of the paper.*
>
> Thank you for pointing this out, will clarify.
>
> **Comment:** *I’m not sure if the term on
> line 85
> is correct.*
>
> We will justify that expression.
>
> Thank you for the remaining grammar/stylistic suggestions; we will incorporate them into the revision.

---

### Official Review · Reviewer_2Z7U · 2021-07-20

**Rating:** 6
**Confidence:** 1

**Summary:**

In this paper, the authors work on bounding the entropy estimate of a discrete variable from the plug-in estimator. The authors rely on their paper last year for density estimation to obtain the entropy estimate.

**Limitations And Societal Impact:**

no need for it

**Main Review:**

The authors prove that the plug-in estimator for discrete density described in Cohen et al 2020 can be used to estimate the entropy of that density. The estimate relies on a well-known result from Information theory described in the second edition of Cover and Thomas book (Equation 2).

The authors also mentioned that:
For fixed d < 1, no technique relying on the plug-in estimator can yield minimax rates [Wu and Yang, 2016]. The plug-in is, however, minimax optimal for d = infinity, [Paninski, 2003] and is among the few methods for which explicitly computable finite-sample risk bound are known.

The authors then say that instead of making an argument about d, they will argue in terms of moments of the distribution to prove that the plug-in estimator achieves minimax rates.

I went over the proofs, and they seem correct to be and I did not thoroughly check them.

Is this sufficient novelty from last year's results? From my perspective, the authors are only adding their results from last year and a minimal change to prove minimax rates. I have the impression that this theoretical paper does not actually provide much insight and adds to the discussion of entropy estimation. Or solves an open question in that field.

I am ok with the paper being presented at the conference. I am just not excited about it.


**Time Spent Reviewing:**

3

---

> ### Author Response · Authors · 2021-08-10
> **We thank the referee for the informative feedback.**
>
> **Comment:** *the authors are only adding their results from last year and a minimal change to prove minimax rates. I have the impression that this theoretical paper does not actually provide much insight and adds to the discussion of entropy estimation*
>
> "results from last year" refers to the paper
> Cohen, Kontorovich, Wolfer. "Learning discrete distributions with infinite support", NeurIPS 2020.
>
> Due the the conference anonymity requirements, we cannot comment on the author overlap between that paper and the present one. Cohen et al. '20 list 5 theorems as their Main Results; the present paper lists 4 theorems. A cursory glance at the statements and proof techniques should convince any reader that there is almost no overlap between the two papers.

---

### Decision · Program_Chairs · 2021-09-27

**Decision:**

Accept (Poster)

**Comment:**

This paper considers the fundamental problem of estimating entropy of discrete distributions and analyzes the error of the empirical estimator for this problem. If one relies on L1 continuity a dependence on the domain size d is unavoidable (there will be a log d term). The authors define a new notion of continuity for entropy that provides nice bounds on the estimation error of empirical distributions. The paper is worth publishing.